# The Effects of Zumba Fitness^®^ on Respiratory Function and Body Composition Parameters: An Eight-Week Intervention in Healthy Inactive Women

**DOI:** 10.3390/ijerph20010314

**Published:** 2022-12-25

**Authors:** Adriana Ljubojevic, Vladimir Jakovljevic, Snezana Bijelic, Ioan Sârbu, Dragoș Ioan Tohănean, Constantin Albină, Dan Iulian Alexe

**Affiliations:** 1Faculty of Physical Education and Sport, University of Banja Luka, 78000 Banja Luka, Bosnia and Herzegovina; 22nd Department of Surgery—Pediatric Surgery and Orthopedics, “Grigore T. Popa” University of Medicine and Pharmacy, 700115 Iasi, Romania; 3Faculty of Physical Education and Mountain Sports, Transilvania University of Brașov, 500036 Brasov, Romania; 4Faculty of Physical Education and Sports, University of Craiova, 200585 Craiova, Romania; 5Faculty of Movement, Sports and Health, Sciences, “Vasile Alecsandri” University of Bacau, 600115 Bacau, Romania

**Keywords:** body fat, vital capacity, dance, fitness, sedentary, health

## Abstract

Background: Women are considered less active than men, and a sedentary lifestyle particularly affects middle-aged women and affects their overall health. Recommending group fitness programs that have a positive effect on women’s overall health is an important strategy of any health policy. Thus, the purpose of this study was to investigate how the Zumba Fitness^®^ workout affects healthy inactive women’s respiratory function and body composition. Methods: The research was conducted for eight weeks on a sample of 41 females aged 35 to 45 divided into two groups: experimental (21 subjects: age 38.52 ± 2.29) and control (20 subjects: age 39.45 ± 3.01). They were tested before, four weeks, and eight weeks after the intervention period. The respiratory functions were tested using spirometry and the body composition parameters by the Tanita body analyzer. The experimental group practiced Zumba Fitness^®^ three times per week for 60 min (24 training sessions in total). The control group was not physically active during the intervention period. Results: After the Zumba Fitness^®^ program, the experimental group showed a significant improvement in the following respiratory function parameters: forced expiratory volume in the first second (M = 4.02 ± 0.69; MD = 0.65, *p* = 0.01, ES = 0.14), vital capacity (M = 4.10 ± 0.65: MD = −0.63, *p* = 0.03, ES = 0.11) and lung age (M = 30.95 ± 10.30; MD = 8.52, *p* = 0.04, ES = 0.10). The body composition parameters were significantly decreased: body mass (M = 60.09 ± 7.57; MD = 6.32, *p* = 0.03, ES = 0.11), body mass index (M = 20.44 ± 2.63; MD = 2.61, *p* = 0.01, ES = 0.13) and fat mass (M = 16.07 ± 4.87; MD = 3.95, *p* = 0.03, ES = 0.11). Conclusions: The results of the current study suggest that the Zumba Fitness^®^ workout is a beneficial exercise method not only for reducing body parameters but also for improving respiratory function in inactive women.

## 1. Introduction

In recent years, great efforts have been made to raise awareness of the negative consequences of hypokinesia on human health and the importance of physical activity in establishing and maintaining overall health [1,2]. Physical inactivity has been identified as the fourth leading risk factor for global mortality (6% of deaths globally). Overall, 39% of adults aged 18 years and over were overweight in 2016 and about 13% of the world’s adult population (11% of men and 15% of women) is obese [3]. By promoting various forms of physical exercise and healthy nutrition, the fitness industry is gaining an increasingly significant role in society. Zumba Fitness^®^ has become a very popular exercise model that combines all elements of fitness: cardio, muscle conditioning, balance, and flexibility [4]. Exercising is based on the principles of aerobic, interval training, and strength exercises, which accelerate the consumption of calories, improve the work of the cardiovascular system, and strengthens the whole body [5]. It represents a combination of aerobic exercise and Latin American dances choreographed to fun Latin music. Fitness programs that include dancing to music encourage women to exercise more consistently [6,7], which is essential for testing the health effect of group fitness programs. Research shows that Zumba Fitness^®^ is an inexpensive and, from the perspective of exercise continuity, a sustainable model of programmed physical exercise to improve health and aesthetic appearance [8]. The results of recent research show positive effects on changes in body composition [5,9,10,11,12]. Some improvements in Health-related Quality of Life [13,14] and Emotional Wellbeing [15] are confirmed in the latest studies on the effect of Zumba Fitness^®^. What makes this concept of Zumba Fitness^®^ particularly interesting is how closely it relates to how women perceive their bodies. Women who practice Zumba lose weight and lower their body mass index (BMI) while feeling better about their bodies overall [16]. All of these aspects of Zumba Fitness^®^ are beneficial for women’s overall health.

However, there is a noticeable lack of research that tests the effects of Zumba Fitness^®^ on the respiratory function of women. Although it represents a cause-and-effect relationship between continuity and exercise, improving respiratory function parameters is treated as a secondary effect. As a result, it receives less attention in scientific studies that examine the outcomes of group fitness programs. The respiratory function of women is known to be influenced by the different phases of the menstrual cycle and by common hormonal and metabolic conditions [17]. Aging specifically reduces the effectiveness of pulmonary gas exchange, which is more of a concern for inactive women [18]. Their capacity to generate increased ventilation during exercise is smaller in comparison to men’s [19]. However, regardless of gender, those who are physically active have better results on vital capacity indicators than those who are not [20]. Several tests have been designed and are regularly used to evaluate the respiratory system’s performance. The most popular technique for evaluating the effects of ongoing exercise is spirometry. It provides diagnoses, determines functional and work capacity, and measures the effects of different kinesiology programs. Moreover, in order to protect women’s health and vitality, vital capacity must be examined because it can be an indication of a variety of diseases, such as obesity and incorrect lung ventilation [21]. The most effective exercise for boosting vital capacity is aerobic exercise [22], which is essentially Zumba Fitness^®^.

Previous research on the effects of Zumba Fitness^®^ has left a scientific gap in the knowledge surrounding its effects on the respiratory function in sedentary inactive women. Therefore, the aim of this research is to examine the effects of Zumba Fitness^®^ on respiratory function and body parameters in healthy inactive women. We hypothesized that eight weeks of Zumba Fitness^®^ would improve respiratory function and body composition parameters by maintaining fat-free mass and reducing total body mass, BMI, and fat mass.

## 2. Materials and Methods

### 2.1. Study Design

Randomly selected women were divided into two groups: an experimental group (Zumba Fitness^®^ intervention) and one control group (no exercise intervention). Both groups were tested at three-time points during the intervention period: (1) at baseline, (2) four weeks after the post-intervention period, and eight weeks after the post-intervention period. The body composition and respiratory function measures in each group were recorded. All participants volunteered in this study and provided informed consent prior to participation. This study was conducted according to the guidelines laid out in the Declaration of Helsinki for research of human subjects, and the protocol was approved by the Ethics Committee of the Faculty of Sport and Physical Education at the University of Banja Luka (11/1.326-1/22).

### 2.2. Participants

The research was conducted on a sample of 41 women aged 35 to 45 employed in the bank administration sector, with more than 6 h of sedentary tasks in their habitual working day. Inclusion criteria state that the participants must be female, physically inactive (less than 120 min of moderate-to-vigorous physical activity per week), inexperienced at Zumba Fitness^®^ or any group-based fitness programs, healthy (not using drugs or any medication that causes water retention) and musculoskeletal injury-free. Women who have overcome severe diseases, such as cancer, stroke, heart attack or muscular illness were excluded from the study. During the intervention period, both groups (CG and EG) practiced their habitual food intake (3–5 meals per day). Additionally, during the intervention period, both groups received two nutritional education sessions with recommendations for how to adopt healthy nutrition habits. The Zumba Fitness^®^ program was done during the months of April and May. The test subjects were divided into two groups: The control group (CG) members had no physical activities (usual activities) during the duration of the experimental program. Women who attended 95 percent of the training sessions during the eight-week duration of the Zumba Fitness^®^ program made up the experimental group (EG). A total of 26 women began the Zumba Fitness^®^ program, and, of those, 21 met the requirements to be counted in the final measurement because they attended regularly. The participants in the experimental group did not engage in any other forms of physical activity during the experimental program. The testing was carried out at the Sports Institute by professors from the Faculty of Physical Education and Sports before and after the Zumba Fitness^®^ program. The temperature of the room was around 20 °C.

### 2.3. Procedures

#### 2.3.1. Body Composition

All body composition parameters were measured using a TANITA BC-418MA III body composition analyzer (Tokyo, Japan). The examinees were tested in sportswear, in the morning hours (between 8.00–9.00 a.m.). The weight of the equipment entered in the device software was 0.300 g. They held electrodes in their hands as they stood barefoot on the lower portion of the body composition analyzer. Their body height and age were entered into the device. The following factors were chosen to investigate how the Zumba Fitness^®^ program affected changes in body composition: body height (BH), body mass (BM), body mass index (BMI), amount of fat tissue in kilograms (FATMASS), fat-free mass in kilograms (FFM) consisting of muscles, bones, tissues and other fat-free mass and basal metabolic rate expressed in calories (BMRKK). All participants received instructions on hydration, sleep, and diet before each measurement. In order to provide valid data and avoid water retention in the body, all participants stated that they did not menstruate during the testing period.

#### 2.3.2. Respiratory Function

The respiratory function assessments were measured using the “Master Lab” spirometric apparatus of the “Jaeger” company (Wurzburg, Germany). “Master Lab” is fully automated and is connected to an electronic computer Epson PC AX. CAP (Computer-Aided Pulmonary Diagnostic) software was installed on the computer, consisting of programs for examining lung functions: spirometry, flow–volume curves, diffusion using the single breath method and body plethysmography. All technical elements, mathematical algorithms, and reference values were predetermined in the programs in accordance with CECA and ATS requirements. The pulmonary function test followed the American Thoracic Society and European Respiratory Society (ATS/ERS) guidelines [23]. The following variables were investigated to see how the Zumba Fitness^®^ program affected changes in respiratory function parameters: VC—lung vital capacity, FEV1—forced exhaled volume in the first second, PEF—maximum expiratory flow rate, LUNGAGE—lung age. Vital capacity (VC) is the amount of air expelled from the lungs by full-capacity expiration after full-capacity inspiration. After taking a full-capacity breath, the examinee exhales all the air as strongly and as quickly as she can. The result is displayed in liters. Forced expiratory volume in the first second (FEV1) is the amount of air that is exhaled during the first second of forced expiration. The procedure is the same as for VC. The value of the obtained result is expressed in liters. Maximum expiratory flow (PEF) is the maximum air velocity registered during a forced exhalation started with a maximum inspiration, and the value is expressed in liters per second. Each variable was measured three times, and the best result was recorded. Lung age (LUNGAGE) shows the physiological age of the lungs based on the tested parameters.

#### 2.3.3. Zumba Interventions 

The eight-week Zumba Fitness^®^ program was completed by each participant in the experimental group. Zumba Fitness^®^ training sessions were held three times a week in the evening, outside working hours, for a total of 24 training sessions. All participants performed the Zumba Fitness^®^ sessions at the same time (6:30–7:30 p.m.). The classes were led by a certified Zumba instructor (ZIN) and mainly followed the choreography from numbers 43 and 45 of the Zumba Fitness^®^ ZIN DVD. Each Zumba Fitness^®^ training session lasted 60 min and contained the basic concept of Zumba Fitness^®^ exercise: warm-up, the main part of the training, cooling down, and stretching. The warm-up lasted 8–10 min (2–3 music tracks, tempo 120–135 bpm) and was performed through basic movements (march, step touch, side to side), with a gradual acceleration of the music tempo without any hops or jumps. The main part of the training session lasted 40–45 min and was performed with the prepared Zumba music (8–10 original Zumba songs), which regulated the change in tempo and dynamics of the performance of dance choreography (tempo 140–180 bpm). The choreography alternated between merengue, reggaeton, salsa, samba, belly dance, cha-cha-cha, tango, and others. The dances lasted between 3–5 min, and the rest intervals between the dances lasted 15–30 s. The tempo of the music and the complexity of the choreography controlled the exercise’s intensity, which fluctuated during the training session. Cooling down, the last component of the workout, lasted for 5 to 10 min and included light dancing moves while listening to relaxing music in order to progressively drop the heart rate and achieve mental and psychological relaxation (tempo up to 100 bpm). There were fewer motions with the hands over the head and the movements were calmer overall. Stretching was done to increase flexibility, as well as to relax muscles and prevent muscle soreness. The Zumba Fitness^®^ program was designed with care to gradually raise exercise intensity as the exerciser’s body adapts to the loads of previous training sessions (load progression method). The intensity of sessions was assessed by the 0 to 10 rating perceived exertion (RPE) Borg scale [24]. The intensity changes during the training session were indicated by the instructor to ensure that no one made strenuous efforts and maintained moderate-to-vigorous-intensity physical activity (6–8 on the Borg scale). At the end of each session, participants declared the average intensity of the session and attendance.

### 2.4. Statistical Analysis

The obtained results were analyzed with the statistical package SPSS 20.0 (IBM, Armonk, NY, USA). The basic descriptive parameters of the initial and final measurements (arithmetic mean—M and standard deviation—SD) were calculated for six variables that tested body composition and for four variables that tested respiratory function. Descriptive parameters were calculated for the complete sample and then individually for the experimental and control groups. The Kolmogorov–Smirnov test was used to determine whether the analyzed results of the initial and final measurements had a normal distribution, and the results were found to be normal (*p* > 0.05). An independent t-test at the level of *p* < 0.05 was used to establish significant differences in body composition and respiratory function between the experimental and control groups at the initial measurement. The univariate analysis of variance (ANOVA) was used to test the statistical significance of the differences within the group at the initial/transit, transit/final, and initial/final measurement for the parameters of body composition and respiratory function. The level of statistical significance was set at *p* < 0.05. All results are presented in tables. In this study, the effect size values were calculated and analyzed using Cohen’s d. Guidelines for the interpretation of this value were also taken based on analyses [25] and were: 0.01 = small effect, 0.065 = moderate effect, 0.14 = large effect.

## 3. Results

Table 1 shows the results of the body composition and respiratory function parameters at the initial measurement for both groups. Based on the standard deviation and the Kolmogorov–Smirnov test, it can be noted that the results in both groups at the initial measurement are homogeneous and normally distributed. Parametric statistics were used in the further analysis and data processing. More precisely, the independent *t*-test was used at the significance level of *p* < 0.05. The obtained values indicate the homogeneity of the test groups before the start of the experimental program. It is expected that the eventual positive effects of the experimental program are shown due to the absence of differences at the initial measurement.

Table 2 shows the results of the differences between the tested initial/transit, transit/final, and initial/final measurements within the experimental and control groups in the women’s respiratory function (*p* < 0.05). The results of the univariate ANOVA reveal statistically significant differences only between the initial and final measurements of the experimental group in the forced exhaled volume in the first second (F = 4.91 *, MD = 0.65 ^*^
*p* = 0.01, ES = 0.14), vital capacity (F = 3.71 *, MD = −0.63 * *p* = 0.03, ES = 0.11) and lung age (F = 3.28 *, MD = 8.52 * *p* = 0.04, ES = 0.10). In the control group, no statistically significant difference was observed in the examined parameters.

Table 3 shows the results of the differences between tested initial/transit, transit/final, and initial/final measurements within the experimental and control groups in the body composition of the women (*p* < 0.05). The results of the univariate ANOVA reveal statistically significant differences only between the initial and final measurements of the experimental group in body mass (F = 3.54 *, MD = 6.32 * *p* = 0.03, ES = 0.11), BMI (F = 4.28 *, MD = 2.61 * *p* = 0.01, ES = 0.13) and fat mass (F = 3.72 *, MD = 3.95 * *p* = 0.03, ES = 0.11). No statistically significant difference in the examined parameters was observed in the control group.

## 4. Discussion

The main results of this study show that an eight-week Zumba Fitness^®^ intervention provided positive changes in the respiratory function and body composition of inactive women. In addition, Zumba Fitness^®^ could be used for the improvement of vital capacity and decreased body fat and body mass index in women. These changes were associated with large effect sizes.

The obtained values of the parameters of the respiratory function at the final measurement indicate that the implemented Zumba Fitness^®^ program had a positive effect on increasing the vital capacity of the lungs (VC) on average by (0.12) or by (3.5%) and the forced exhaled air volume in 1 s (FEV1) on average by (0.17) or by (5.4%). It is supposed that continuous Zumba Fitness^®^ exercise caused better ventilation in the respiratory system and “cleaning” of the respiratory organs and pathways. The lungs do not have the ability to decrease or increase their volume. They are activated as a result of changes in the ratio between internal and external air pressures. That stems from a reduction or expansion of the chest. The normal movement of the joints that are connected to the chest are important for allowing optimal respiration ranges. There is evidence that stretching respiratory muscles increases the capacity for chest wall expansion, suggesting an improvement in lung ventilation [26]. Some research suggests that an aerobic training program combined with respiratory muscle stretching increases the functional and ventilatory capacities [27]. The upper-body strength and stretch exercises from the Zumba Fitness^®^ program probably contributed to a better efficiency of the intercostal muscles in the activities of increasing the volume of the chest, which was consequently reflected in better test results at the final measurement. Twist positions also have a positive effect on the elasticity of the chest. Rotations, such as in Zumba Fitness^®^ choreography, may improve respiratory functions in all trainees, especially those with incorrect posture [28]. It is considered that the programmed and controlled increase in exercise intensity contributed to the adequate adaptation of the functions of the respiratory system. This resulted in a decrease in the value of the variable that analyzed the age of the lungs (LUNGAGE) on average by (5.05), i.e., by (128%). At the same time, the control group did not show any changes in results compared to the initial measurement.

Research on the aerobic capacity of women conducted by Barrene et al. [29] indicates that a 12-week Zumba fitness program outside of working hours significantly improved the VO2 peak (5%; *p* = 0.02) and decreased heart rate during 100 W cycle exercise (−7 bpm; *p* = 0.01). Similar results attained by Domene et al. [10] (by 3.1 mL/kg^−1^/min^−1^, *p* = 0.05; effect size 0.56) and Delextrat et al. [30] (+3.6%, *p* = 0.008; moderate effect size) show an increasing maximal oxygen uptake after only eight weeks of Zumba Fitness^®^ intervention. Donath et al. [31] claim that Zumba training can be applied to improve aerobic fitness measured by the six min walking test (INT: +21%. CON: −2%, *p* < 0.001, effect size 0.83). Furthermore, recent research [13] shows statistical improvements in the cardiorespiratory fitness test scores (time in the 2 km test, as well as in the estimated VO2 max) after 16 weeks of Zumba fitness^®^ intervention. A meta-analysis of the effects of Zumba Fitness^®^ on improving maximum oxygen consumption [6] indicates that the younger population adapts to exercise much faster than the older. Namely, research conducted on a sample of overweight postmenopausal women [14] shows that the 12-week Zumba program did not improve maximum oxygen consumption (VO2 peak). Improvements in cardiorespiratory fitness could be caused by exercise intensity and adequate training progression during the intervention. The Zumba Fitness^®^ classes were led by a certified instructor (ZIN) following progressive training based on the original structure of Zumba Fitness^®^. This apparently provides better organization and control within the classes, possibly related to significant changes in cardiovascular outcomes. Zumba Fitness^®^ is an activity of a primarily aerobic nature, and improvements in respiratory function can be expected if the program is adequately designed and if it lasts long enough to contribute to the body’s adaptation. Although the findings are promising, comparing results is difficult since most of the presented studies used field tests and different procedures for aerobic capacity measurements. The respiratory function conducted in this research was indirectly measured in laboratory conditions in the resting mode, which is not the case with the above-mentioned research. However, the results of the conducted research indicate that non-field laboratory tests of respiratory functions in women could be an effective method for future assessments of the effects of Zumba Fitness^®^. Furthermore, the lack of research on the effects of Zumba Fitness^®^ on vital capacity precludes a more objective comparison of the obtained results. In general, the small amount of research that investigates the effects of exercising on respiratory function, especially in inactive women, can be explained by the fact that the primary motive of group fitness exercise is to change physical appearance [32].

Furthermore, the experimental group experienced significant changes in body composition parameters after Zumba Fitness^®^ classes: there was a decrease in the weight of subcutaneous fat tissue (by 1 kg or 5% on average), which directly resulted in a decrease in body mass (on average by 0.83 kg or by 1.2%) and body mass index (on average by 0.39 or by 1.7%). The implemented Zumba Fitness^®^ program reduced fatty tissue, which interferes with the normal functioning of the body and negatively affects women’s health as a whole.

The results of previous research show that the effects of Zumba fitness^®^ exercise on body composition parameters vary concerning the sample of respondents. The greatest effects on the reduction of subcutaneous fat tissue and BMI were shown in the population of overweight and obese women [9,10]. Although significant progress has been noticed in the population of healthy recreational women [33,34], it should be kept in mind that the effectiveness of the Zumba Fitness^®^ program is conditioned by the length of its duration. Research that was conducted for 4–6 weeks did not show significant effects on changes in body composition after the Zumba Fitness^®^ program [35], as in this research. Most of the research conducted for eight weeks indicates the positive effects of the Zumba Fitness^®^ program on the reduction of subcutaneous fat tissue and BMI [12,36]. In particular, the Zumba Fitness^®^ program that is implemented for a longer duration of 12 weeks can lead to an increase in total muscle mass [37]. Fat-Free Mass describes all of the tissues in the body that are not adipose (fat) tissue. In the research conducted here, there is no improvement in Fat-Free Mass (*p* = 0.5; effect size 0.02). In addition to lasting eight weeks, the applied Zumba Fitness^®^ program did not include the use of props (dumbbells, weight discs, elastic bands, etc.) for greater activation of muscle mass. The basal metabolic rate is greatly affected by the number of muscles engaged. The absence of an increase in muscle mass caused the absence of changes in basal metabolic rate (*p* = 0.84; effect size 0.00).

Future research on the effects of Zumba fitness^®^ in inactive women could include external load elements (Zumba toning^®^ with dumbbells and barbells) to examine the effect on increasing muscle mass. Furthermore, monitoring changes in smoking or dietary habits could reveal additional benefits of this type of exercising. However, some limitations must be recognized in the present study. First, the study sample was small and limited to inactive adult women. These findings may not be generalizable to other populations with different characteristics. Secondly, the study has a relatively short intervention period, which limited the outcomes of a possible effect of a longer period of practicing Zumba Fitness^®^. Furthermore, to achieve the object of exercise intensity, a pulse meter (heart rate measurement) could have been used, but instead, we used the Borg scale as a valid method.

## 5. Conclusions

The current study suggests that an eight-week Zumba Fitness^®^ exercise program is beneficial for improving the respiratory function and body parameters in inactive women. The improvements were especially generated in increasing the vital capacity and forced expiratory volume in the first second, reducing body mass, adipose tissue and body mass index and reducing the estimation of the physiological age of the lungs. A comprehensive approach to examining the effects of Zumba Fitness^®^ indicates its great benefit to women’s overall health. Furthermore, a large percentage (95%) of the surveyed women who completed the experimental program indicated that fun and challenging dance-structured exercises such as Zumba Fitnes^®^ could provide the sustainability of this form of group fitness training. 

## Figures and Tables

**Table 1 ijerph-20-00314-t001:** The difference between the experimental and control groups at the initial measurement.

					Statistical Difference between Groups		
Variable	Group	N	Mean	SD	t	df	Sig.
Body parameters							
Age	CG	20	39.45	3.01	0.47	39	0.22
	EG	21	38.52	2.29			
BM	CG	20	62.58	7 ± 0.57	−1.48	39	0.14
	EG	21	65.94	6.87			
BMI	CG	20	22.25	2.77	−0.92	39	0.35
	EG	21	23.05	2.76			
BMRKK	CG	20	1386.80	92.59	−0.92	39	0.36
	EG	21	1417.00	114.70			
FAT MASS	CG	20	16.99	5.41	−1.82	39	0.07
	EG	21	19.96	5.00			
FFM	CG	20	45.61	3.49	−0.69	39	0.48
	EG	21	46.40	3.74			
Respiratory function							
FEV1	CG	20	3.11	0.45	−1.39	39	0.17
	EG	21	3.30	0.40			
VC	CG	20	3.49	0.39	−0.31	39	0.75
	EG	21	3.45	0.40			
PEF	CG	20	288.25	90.71	−1.42	39	0.16
	EG	21	327.95	87.13			
LUNGAGE	CG	20	34.85	10.76	−0.28	39	0.78
	EG	21	35.80	11.05			

Legend: CG—control group, EG—experimental group, BM—body mass (kg), BMI—body mass index, BMRKK—kilocalorie basal metabolic rate, FATMASS—total fat mass, FFM—fat-free mass, FEV1—forced exhaled volume in the first second, VC—lung vital capacity, PEF—maximum forced expiratory flow when the lungs are maximally filled with air, LUNGAGE—lung age, M—arithmetic mean, SD—standard deviation.

**Table 2 ijerph-20-00314-t002:** Difference between measurements (initial, transit, and final) within groups (EG and CG group) for respiratory function.

Variable	Group	ANOVA						Measure	Multiple Comparisons	
		In M ± SD	Tr M ± SD	Fn M ± SD	F	Sig	ES		Mean Diff.	Sig.
FEV1	EG	3.30 ± 0.40	3.71 ± 0.79	4.02 ± 0.69	4.91	0.01	0.14	in-tr	−0.33	0.33
								tr-fn	−0.31	0.41
								in-fn	0.65	0.01
	CG	3.11 ± 0.45	3.45 ± 0.56	3.51 ± 0.65	2.88	0.06	0.09	in-tr	0.34	0.19
								tr-fn	−0.06	1.00
								in-fin	−0.40	0.09
VC	EG	3.47 ± 0.51	3.80 ± 0.98	4.10 ± 0.65	3.71	0.03	0.11	in-tr	0.33	0.45
								tr-fn	−0.29	0.63
								in-fn	−0.63	0.03
	CG	3.49 ± 0.39	4.00 ± 1.25	4.09 ± 0.75	2.72	0.07	0.09	in-tr	−0.51	0.22
								tr-fn	−0.09	1.00
								in-fn	−0.60	0.10
PEF	EG	295.88 ± 0.81.54	327.95 ± 87.13	355.19 ± 81.50	2.71	0.07	0.08	in-tr	−32.67	0.63
								tr-fn	−27.24	0.88
								in-fn	−59.90	0.07
	CG	288.25 ± 90.71	336.45 ± 59.96	340.15 ± 90.19	2.52	0.09	0.08	in-tr	−48.20	0.20
								tr-fn	−3.70	1.00
								in-fn	−51.90	0.15
LUNG AGE	EG	39.47 ± 11.03	35.80 ± 11.07	30.95 ± 10.30	3.28	0.04	0.10	in-tr	3.67	0.83
								tr-fn	4.86	0.45
								in-fn	8.52	0.04
	CG	34.85 ± 10.76	33.45 ± 11.23	33.35 ± 10.93	0.11	0.89	0.00	in-tr	1.40	1.00
								tr-fn	0.10	1.00
								in-fn	1.50	1.00

Legend: CG—control group. EG—experimental group. In—initial, tr—transit, fn—final measurement, M—arithmetic mean, SD—standard deviation. The statistical difference between the initial and transit, and the initial and final measurements.

**Table 3 ijerph-20-00314-t003:** Difference between measurements (initial, transit, and final) within groups (EG and CG) for body parameters.

Variable	Group	ANOVA						Measure	Multiple Comparison	
		In M ± SD	Tr M ± SD	Fn M ± SD	F	Sig.	ES		Mean Diff.	Sig.
BM	EG	65.94 ± 6.87	63.61 ± 7.81	60.09 ± 7.57	3.54	0.03	0.11	in-tr	2.80	0.73
								tr-fn	3.52	0.43
								in-fn	6.32	0.03
	CG	62.58 ± 7.57	61.40 ± 8.82	60.10 ± 7.66	0.47	0.62	0.02	in-tr	1.18	1.00
								tr-fn	1.30	1.00
								in-fn	2.48	1.00
BMI	EG	23.05 ± 2.76	21.72 ± 3.21	20.44 ± 2.63	4.28	0.01	0.13	in-tr	1.32	0.43
								tr-fn	1.28	0.46
								in-fn	2.61	0.01
	CG	22.25 ± 2.77	20.82 ± 3.27	22.87 ± 6.32	1.13	0.33	0.04	in-tr	1.43	0.93
								tr-fn	−2.04	0.44
								in-fn	−0.62	1.00
BMRKK	EG	1417.61 ± 114.48	1460.09 ± 127.77	1502.28 ± 116.12	2.71	0.07	0.08	in-tr	−42.47	0.74
								tr-fn	−42.19	0.75
								in-fn	−84.66	0.07
	CG	1386.80 ± 92.59	1444.45 ± 144.16	1417.75 ± 93.65	1.66	0.20	0.06	in-tr	−57.65	0.33
								tr-fn	2.70	1.00
								in-fn	−54.95	0.38
FAT MASS	EG	20.02 ± 4.95	18.30 ± 4.27	16.07 ± 4.87	3.72	0.03	0.11	in-tr	1.72	0.73
								tr-fn	2.23	0.39
								in-fn	3.95	0.03
	CG	16.99 ± 5.41	17.45 ± 4.33	16.13 ± 5.24	0.35	0.70	0.01	in-tr	−0.46	1.00
								tr-fn	1.32	1.00
								in-fn	0.86	1.00
FFM	EG	46.40 ± 3.74	47.97 ± 3.97	49.32 ± 4.86	2.52	0.09	0.08	in-tr	−1.58	0.70
								tr-fn	−1.35	0.91
								in-fn	−2.92	0.09
	CG	45.61 ± 3.49	47.82 ± 4.34	45.64 ± 3.25	2.30	0.11	0.07	in-tr	−2.21	0.20
								tr-fn	2.18	0.21
								in-fn	−0.04	1.00

Legend: CG—control group. EG—experimental group. In—initial, tr—transit, fn—final measurement, M—arithmetic mean, SD—standard deviation. The statistical difference between the initial and transit, transit and final, and initial and final measurements.

## Data Availability

The data presented in this study are available on request from the corresponding author. The data are not publicly available due to ethical restrictions around participant privacy.

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
