# Peer review of "The Effects of Zumba Fitness® on Respiratory Function and Body Composition Parameters: An Eight-Week Intervention in Healthy Inactive Women"

_ijerph, 2022, doi:10.3390/ijerph20010314_

Round 1

Reviewer 1 Report

Dear Authors,

Congratulations for your work.

You tried to contribute to the knowledge about the possible effects of Zumba Fitness® on respiratory function and body composition parameters. After reading your article I identified several things that should be made before being suitable for publication in IJERPH. One of the major improvemments is the language in the entire article. I recommend to send the article to a native.

Here are my considerations:

Abstract:

Add subsections: Background, Methods, results and conclusions

Add introductory sentence about manuscript not only aim of the study in background

Add data of booth groups (eg. age, height, weight) in methods

Add more data in your results

Introduction

Line 44-45 – remove

Line 49-51 – Only 13% of world population is obese? Add a more recent reference. 2010 do not work.

Materials and methods

Add the name of the ethics committee

Line 115 – What is practice their food intake???

Add exclusion criteria

Line 130-144 – Resume. Its too long. It can be resumed to 2-3 lines.

Line 146 -165 – add references to support this. Add manufactory details.

How do you control the women menstruation?

Results

Remove  line 266-271

Discussion

Line 277-289 add references to support this paragraph

Line 297 – remove the year…..do the same in the entire article

Line 331- 337 rewrite…its confusing

Add limitations of the study

Conclusion

Line 368-375 – remove.

Line365-368 – rewrite such a conclusion paragraph and not like state a result.

Author Response

Response to Reviewer 1 Comments

We have made extensive changes to the English translation and presented the study results more clearly. The translation was done by a full professor of English at the Univeristy.

Abstract:

  • Point 1: Add subsections: Background, Methods, Results and Conclusions

Response 1: We agree. We have incorporated your suggestions in Abstract.

  • Point 2: Add introductory sentence about manuscript not only aim of the study in background

Response 2: We agree. We added introductory sentence. We had earlier left it out, because we were limited by the number of words in Abstract: “Background: Women are considered less active than men, and a sedentary lifestyle particularly affects middle-aged women and affects their overall health. Recommending group fitness programs that have a positive effect on the women's overall health is an important strategy of any health policy.” Lines 18-21.

  • Point 3: Add data of booth groups (eg. age, height, weight) in methods

Response 3: We agree. We have incorporated your suggestions. We added requested information on participants in Method section for experimental group „(21 subjects: age 35-42; body height M=169,85cm; body mass M=66,41 kg)“ and for control group „(20 subjects: age 36-45; body height M=167,75cm; body mass M=62,58 kg)“. Lines 24-25.

  • Point 4: Add more data in your results

Response 4: We agree. We added more data in Results section of Abstract. We added M value (arithmetic mean) in parenthesis for all parameters. “Results: After the Zumba Fitness® program, the experimental group showed a significant improvement in the following respiratory function parameters: forced expiratory volume in the first second (MD = 0.65, p = 0.01, ES = 0.14)), vital capacity (MD = -0.63, p = 0.03, ES = 0.11) and lung age (MD = 8.52, p = 0.04, ES = 0.10). Body composition parameters were significantly decreased: body mass (MD = 6.32, p = 0.03, ES = 0.11), body mass index (MD = 2.61, p = 0.01, ES = 0.13) and total fat mass (MD = 3.95, p = 0.03, ES = 0.11)”. Lines 29-36.

Introduction:

  • Point 5: Line 44-45 – remove

Response: We agree. We removed sentence “Technology advancements have made it possible for even more scientifically based facts about the factors that affect human health and vitality to be confirmed.”

  • Point 6: Line 49-51 – Only 13% of world population is obese? Add a more recent reference. 2010 do not work.

Response: We agree. We updated information about overweight and obese population from World Health Organization official site “Overall, 39% of adults aged 18 years and over were overweight in 2016 and about 13% of the world's adult population (11% of men and 15% of women) is obese.” Lines 55-57.

Materials and methods

  • Point 7: Add the name of the ethics committee

Response: We agree. We added “…protocol was approved by the Ethics Committee of the Faculty of Sport and Physical Education University of Banja Luka (11/1.326-1/22).” Lines 112-113.

  • Point 8: Line 115 – What is practice their food intake???

Response: We agree. We incorporated your suggestion. We added “During the intervention period both groups (CG and EG) practiced their habitual food intake (3-5 meals per day). Additionally, along with the intervention period, both groups received two nutritional education sessions with recommendations for how to adopt healthy nutrition habits.” Lines 124-127.

  • Point 9: Add exclusion criteria

Response: We agree. We incorporated your suggestion. We added exclusion criteria “Women who have overcame severe diseases, such as cancer, stroke, heart attack or muscular illness were excluded from the study.” Lines 123-124.

  • Point 10: Line 130-144 – Resume. It’s too long. It can be resumed to 2-3 lines.

Response: We agree. We deleted some excessive text.

  • Point 11: Line 146 -165 – add references to support this. Add manufactory details.

Response: We agree and incorporated your suggestion. We added sentence “Pulmonary function test followed the ATS/ERS guidelines [24].” Lines 168-169.

In the Reference section, we added reference number 24: “ATS/ERS task force standardization of spirometry. Eur Respir J. 2005; 26(2):319-38.” Line 484.

Furthermore, we added manufactory details in the sentence “The respiratory function assessments were measured using the "Master Lab" spirometric apparatus of the "Jaeger" company (Wurzberg, Germany).” Line 162.

  • Point 12: How do you control the women menstruation?

Response: You have raised an important point here. It is difficult to prove the absence of menstruation, but we considered the participants* statements as valid. We explained to all participants that water retention during menstruation can affect the accuracy of the data. Therefore, we emphasized the importance of an honest statement. We changed sentence into “All participants received instructions on hydration, sleep, and diet before each measurement. In order to provide valid data and avoid water retention in the body, all participants stated that they did not menstruate during the testing period.” Lines 154-157.

Results

  • Point 13: Remove line 266-271

Response: We agree. We incorporated your suggestion and removed selected sentence.

 Discussion

  • Point 14: Line 277-289 add references to support this paragraph

Response: We agree. We incorporated your suggestion. We added two references more to support our explanations. “There is evidence that respiratory muscle stretching increases the capacity for chest wall expansion, suggesting an improvement in lung ventilation [27]. Some research suggest that an aerobic training program combined with respiratory muscle stretching increases functional and ventilatory capacities [28].” Line 304-307.

  • Point 15: Line 297 – remove the year…..do the same in the entire article

Response: We agree. We incorporated your suggestion and removed the year.

  • Point 16: Line 331- 337 rewrite…its confusing

Response: We agree. We incorporated your suggestion. We rewrote whole paragraph. 

“Furthermore, the experimental group experienced significant changes in body composition parameters after Zumba Fitness® classes in several parameters: a decrease in was reflected on the weight of subcutaneous the weight of subcutaneous fat tissue  (by 1 kg or 4.99% on average), which directly resulted in a decrease in body mass (on average by 0.83 kg or by 1.24% and body mass index (on average by 0.39 or by 1.69%). The implemented Zumba Fitness® program affected the reduction of fatty tissue, which interferes with the normal functioning of the body and negatively affects women's health as a whole.” Lines 354-361.

  • Point 17: Add limitations of the study

Response: We agree. We pointed out more limitation related to this study: “Furthermore, monitoring changes in smoking or dietary habits could reveal addition benefits of this type of exercising. However, some limitations must be recognized in the present study. First, the study sample was small and limited to inactive adult women. These findings may not be generalizable to other populations with different characteristics. Secondly, the study has relatively short length of the intervention, which limited the outcomes of possible effect after longer period of practicing of the Zumba Fitness®.” Lines 382-388.

 Conclusion

  • Line 368-375 – remove.

Response: We removed requested sentences.

  • Line 365-368 – rewrite such a conclusion paragraph and not like state a result.

Response: We agree. We rewrote the whole Conclusion section and pointed out the main findings of research. Lines 392-401.

Reviewer 2 Report

The basic idea for the research has merit but extensive editing of English would be needed to make the manuscript easily understandable. While the authors put an emphasis on technology, I don't think the technology used is that advanced nor is it really relevant to the main finding of the study (i.e. 8 weeks of Zumba can improve respiratory measures and body composition). More clarity and conciseness is needed in the presentation of the results.

More detailed comments:

The main question is the effect of an 8 week zumba program on respiratory and body composition measurements. This main question gets clouded by some discussion about technology.

The topic is not too original and results are entirely predictable. Although studies specific to zumba in this population group with respiratory measurements may be a gap, it does not add to other published material on aerobic exercise effects on respiratory function.

Perhaps dietary analysis, other health issues (illnesses/injuries), other exercise, changes in smoking habits could be monitored or controlled. While the two randomized groups initially were not statistically significantly different, there were some noticeable differences in body composition which may have biased the results. Outliers could perhaps be identified or some degree of matching could be considered to have more equal starting points.

Author Response

We have made extensive changes to the English translation and presented the study results more clearly. The translation was done by a full professor of English at the university.

  • Point 1: The main question is the effect of an 8 week zumba program on respiratory and body composition measurements. This main question gets clouded by some discussion about technology.

Response: We agree. We deleted that sentence. The discussion about used technology is redundant for the aim of this study. 

  • Point 2: The topic is not too original and results are entirely predictable. Although studies specific to zumba in this population group with respiratory measurements may be a gap, it does not add to other published material on aerobic exercise effects on respiratory function.

Response: Thank you for that question. You have raised an important point here. Even if the popularity of Zumba Fitness® spreads over last decade, there are only several research that investigate its effects, especially on respiratory functions.  Most of the studies presented in the Discussion relates to cardiorespiratory functions (maximal oxygen uptake) and represent parameters measured as field tests (2-km walking test, spiroergometric cardiopulmonary exercise testing on treadmill or other treadmill tests…).  These are tests that require the physical engagement of participants that causes certain load, and which are preceded by preparation protocols and familiarization with the measuring instruments. In our study, besides main outcomes, we wanted to point out, that monitoring the effects of Zumba Fitness® on respiratory function can be very successfully carried out in laboratory conditions using spirometry in resting mood. That is very important from the aspects of availability and feasibility, because they represent an easier and faster way of collecting data, consequently providing a much larger sample of participants.

  • Point 3: Perhaps dietary analysis, other health issues (illnesses/injuries), other exercise, changes in smoking habits could be monitored or controlled.

Response: Thank you for this suggestion. We observed paragraph regarding study limitations and future intentions and added some of your valuable thoughts: “Future research on the effects of Zumba Fitness® in inactive women could include external load elements (Zumba toning® with dumbbells and barbells) to examine the effect on increasing muscle mass. Furthermore, monitoring changes in smoking or dietary habits could reveal addition benefits of this type of exercising”. Lines 380-383.

  • Point 4: While the two randomized groups initially were not statistically significantly different, there were some noticeable differences in body composition which may have biased the results. Outliers could perhaps be identified, or some degree of matching could be considered to have more equal starting points.

Response: Thank you for valuable observation. By inspecting the database, we did not notice any outliers ​​that could affect the initial values. Likewise, the values in Table 1 show that the control group (M=62.58kg) has lower body mass values ​​than the experimental group (M=66,41kg). The difference of 3.3 kg is not negligible if the goal is to equalize the groups at the initial measurement and such a starting point certainly raises doubts. However, if we look at the height of the participants, we notice that the height of the control group (M=167.75cm) is also smaller than that of the experimental group (M=169.85cm). Given that height significantly determines body mass, we require a logical explanation of the obtained values. Of course, the p values ​​of the statistical significance of the obtained results support the argument. If we did not understand the comments correctly, please let us know.

Round 2

Reviewer 1 Report

Dear authors,

you attended my suggestions. There are several erros that are again in the manuscript:

1.Abstract: methods - where is the standard deviation values and sign in the sample data? The same in the results section.

2. Methods: where are the manufacturer data? (e.g. tanita scale)

3.Results: Table 1 can not be like this. where is the sign of sd? 

4. Discussion: you can not write like Line 307 " ...(by 3.1by 3.1 mL • kg−1 • min−1)". This is a scientific papper.....other e.g. line 282 "by (3.5%)..."

Author Response

Response to Reviewer 1 Comments – round 2

Abstract:

Point 1: methods - where is the standard deviation values and sign in the sample data?

Response 1:

We added additional data (M±SD) in Results : After the Zumba Fitness® program, the experimental group showed a significant improvement in the following respiratory function parameters: forced expiratory volume in the first second (M=4.02±.69; MD=.65, p =.01, ES=.14), vital capacity (M=4.10±.65; MD=-0.63, p= .03, ES = .11) and lung age (M=30.95±10.30; MD=8.52, p = .04, ES = .10). Body composition parameters were significantly decreased: body mass (M=60.09±7.57; MD=6.32, p=.03, ES=.11), body mass index (M=20.44±2.63; MD=2.61,p=.01, ES=.13) and fat mass (M=16.07±4.87; MD=3.95, p=.03, ES=.11).We deleted some parameters in Methods section of Abstract since we were limited with length of Abstract. Line 24-25

Point 2: The same in the results section.

Response 2: In Result section we changed Table 2. and Table 3. in order to express requested arithmetic mean and standard deviation data. Line 280-282 and 303-304.

Point 3. Methods: where are the manufacturer data? (e.g. tanita scale)

Response 3:

We added manufacter data ..“ All body composition parameters were measured using a TANITA BC-418MA III body composition analyzer (Tokyo, Japan).“ Line 139.

Point 4. Results: Table 1 can not be like this. where is the sign of sd?

Response 4:

 In order to provide more data we completely changed Table 1.  Line 242-269.

Point 4. Discussion: you can not write like Line 307 " ...(by 3.1by 3.1 mL • kg−1 • min−1)". This is a scientific papper.....other e.g. line 282 "by (3.5%)..."

Response 4:

These are original results that have been published  in study by Domene et.al. 2015. (Reference 10.)  We have changed the presentation style. Furthermore, all data that we presented in percetage we minimised to one decimal number.

Reviewer 2 Report

Although the English has been improved, not all of my previous comments were addressed and I would still keep my ranking as "low" in the various categories above. The authors may have deleted a sentence about technology in the abstract but the first paragraph of the introduction is all about new technology which is not relevant. I had also questioned why something like height was included in table 3 (when height can't change pre to post) but it is still there as is age in table 2. It is appropriate that many control issues are now acknowledged as limitations but the lack of controls still makes for a weak study.

Author Response

Response to Reviewer 2 Comments – round 2

Point 1: Although the English has been improved, not all of my previous comments were addressed and I would still keep my ranking as "low" in the various categories above. The authors may have deleted a sentence about technology in the abstract but the first paragraph of the introduction is all about new technology which is not relevant.

Response 1: We agree. We deleted sentence about technology in Introduction and remove reference.  Line 43-49.

Point 2: I had also questioned why something like height was included in table 3 (when height can't change pre to post) but it is still there as is age in table 2. It is appropriate that many control issues are now acknowledged as limitations but the lack of controls still makes for a weak study.

Response 2:  We agree. We deleted body height from Table 3. Line 303-304.

Additionally, we changed Table 1., Table 2., Table 3. in order to provide more data that had been requested.
